# Experimental and Machine Learning Studies on Chitosan-Polyacrylamide Copolymers for Selective Separation of Metal Sulfides in the Froth Flotation Process

**Keitumetse Monyake [1,*], Taihao Han [2], Danish Ali [1], Lana Alagha [1,3,*] and Aditya Kumar [2,*]**

1   Department of Mining and Explosives Engineering, Missouri University of Science and Technology, Rolla, MO 65409, USA
2   Department of Materials Science and Engineering, Missouri University of Science and Technology, Rolla, MO 65409, USA
3   Thomas J. O'Keefe Institute of Sustainable Supply of Strategic Minerals, Missouri University of Science and Technology, Rolla, MO 65409, USA
*   Correspondence: kcm29c@umsystem.edu (K.M.); alaghal@mst.edu (L.A.); kumarad@mst.edu (A.K.); Tel.: +1-573-612-9306 (K.M.); +1-573-341-6287 (L.A.); +1-573-341-6994 (A.K.)

**Abstract:** The froth flotation process is extensively used for the selective separation of valuable base metal sulfides from uneconomic associated minerals. However, in this complex multiphase process, various parameters need to be optimized to ensure separation selectivity and peak performance. In this study, two machine learning (ML) models, artificial neural network (ANN) and random forests (RF), were used to predict the efficiency of in-house synthesized chitosan-polyacrylamide copolymers (C-PAMs) in the depression of iron sulfide minerals (i.e., pyrite) while valuable base metal sulfides (i.e., galena and chalcopyrite) were floated using nine flotation variables as inputs to the models. The prediction performance of the models was rigorously evaluated based on the coefficient of determination ($R^2$) and the root-mean-square error (RMSE). The results showed that the RF model was able to produce high-fidelity predictions of the depression of pyrite once thoroughly trained as compared to ANN. With the RF model, the overall $R^2$ and RMSE values were 0.88 and 4.38 for the training phase, respectively, and $R^2$ of 0.90 and RMSE of 3.78 for the testing phase. As for the ANN, during the training phase, the overall $R^2$ and RMSE were 0.76 and 4.75, respectively, and during the testing phase, the $R^2$ and RMSE were 0.65 and 5.42, respectively. Additionally, fundamental investigations on the surface chemistry of C-PAMs at the mineral–water interface were conducted to give fundamental insights into the behavior of different metal sulfides during the flotation process. C-PAM was found to strongly adsorb on pyrite as compared to galena and chalcopyrite through zeta potential, X-ray photoelectron spectroscopy (XPS), and adsorption density measurements. XPS tests suggested that the adsorption mechanism of C-PAM on pyrite was through chemisorption of the amine and amide groups of the polymer.

**Keywords:** chitosan; froth flotation; metallic sulfides; machine learning; adsorption mechanism; X-ray photoelectron spectroscopy; zeta potential; total organic carbon



## 1. Introduction

Froth flotation is a highly complex multiphase process that is used to selectively separate minerals based on differences in surface characteristics. Froth flotation is the major separation technique that is used to separate base metal sulfides from the associated uneconomic minerals, including iron sulfide minerals. In this practice, valuable sulfides are commonly floated together as bulk concentrates during the first stages of the process (bulk flotation,) followed by several cleaning stages to selectively separate individual minerals (selective flotation).

The froth flotation process is impacted by a large number of interactive variables [1]. For example, airflow rate affects bubble sizes which subsequently impact the mineral-bubble attachment, gas hold-up, froth depth, etc. [2]. The variables that affect the froth flotation process are divided into three major groups [3,4]: (a) Feed attributes, such as particle size distribution, mass flowrate, mineral specific gravity, liberation, composition, etc.; (b) Physicochemical influences including reagent types and interaction, reagents dosages and addition sequence, water quality, slurry pH, and temperature; and (c) Hydrodynamic aspects including flotation circuit design, type of flotation cell, airflow rate, and froth stability. The flotation performance is usually evaluated based on the recovery of valuable minerals in the flotation concentrates and the quality of the concentrates (value metal grades) [5]. These variables affect the flotation outcomes independently, but their interdependence complicates the process control. Thus, it is essential to develop high-fidelity control systems that can produce quality outcomes of the process given the diverse process variables.

Conventional modeling tools, such as mathematical and statistical approaches, have been used to predict flotation performance through experiments [5–7]. However, such modeling tools have proven to lack the capability to accurately predict the performance of complex systems that have interdependent input variables [5]. For example, Monyake and Alagha [8] recently used response surface methodology (RSM) to optimize pyrite depression in the bulk flotation of chalcopyrite and galena. However, when developing the quadratic equations for response predictions, some insignificant terms had to be excluded as a way of improving the models' accuracy. As a result, some combinations of the eliminated terms could not be studied. Additionally, the correlation coefficient ($R^2$) values of copper recovery and grades were low when using RSM, which agrees with the findings Pilkington et al. [9], where the statistical $R^2$ values for the prediction of artemisinin extraction from Artemisia annua were much lower when compared to artificial neural network (ANN) predictions. This suggested that machine learning (ML) models are superior to RSM at response predictions. Reflecting on these intricacies and the aforementioned insufficiencies of classical modeling approaches, researchers have focused on developing and employing ML models—both supervised and unsupervised—for prediction, and in some cases optimization, of flotation processes [10–16]. When properly trained with high-quality datasets, ML models can be very effective at revealing underlying relationships between experimental process parameters and outputs (herein metal grades and recoveries) and performance predictions. Researchers have applied ML models for prediction of froth flotation outcomes in a wide range of applications. The flotation of a paper de-inking process was studied by Labidi et al. [12] using ANN. Their work showed that ANN was reliable at accurately reproducing the experimental results with high $R^2$ values. Ali et al. [17] assessed the performance of five different ML models [ANN, random forest (RF), adaptive neuro-fuzzy inference system (ANFIS), Mamdani fuzzy logic (MFL), and hybrid neural fuzzy inference system (HyFIS)] to predict the coal ash content in the flotation of fine coal. Their work showed that all ML models performed well with MFL, producing the best prediction performance with $R^2$ of 0.92 on the testing dataset. Khodakarami et al. [10] used a cascade forward neural network with a backpropagation algorithm to predict the impact of five operational parameters in the flotation of fine coal using hybrid polymeric nanoparticles. Nakhaei et al. [15] used ANN to predict the grades and recoveries of copper and molybdenum in the pilot plant column flotation concentrate using the backpropagation model. Their model was also tested at an industrial flotation plant using 92 different datasets collected at different operational conditions and showed good accuracy for the prediction of copper and molybdenum recoveries and grades with $R^2$ values ranging from 0.92 to 0.94. Shahbazi et al. [18] used RF together with its associated variable importance measurements (VIMs) to predict the flotation responses as a function of particle characteristics and hydrodynamic conditions in quartz flotation. The modeling results indicated that the RF model was satisfactorily making predictions with $R^2$ values of 0.96–0.97. An innovative hybrid ML model using a combination of random forest and firefly algorithm (RF-FFA) was developed by Cook et al. to predict the flotation efficiency

of metal sulfides in the bulk flotation of galena and chalcopyrite [19]. The RF-FFA model outperformed other standalone models as it showed high prediction accuracy.

In this study, two ML models (RF and ANN) were developed, trained, and validated for predicting the grades and recoveries of base metals and iron (Pb, Cu, Zn, and Fe) in the concentrates produced from the bulk flotation of galena (PbS) and chalcopyrite ($CuFeS_2$) from a complex sulfide ore of Mississippi Valley Type (MVT) that also contained sphalerite (ZnS) and pyrite ($FeS_2$). The common industrial flotation practice for MVT is to recover galena and chalcopyrite as a bulk float during the first stages while depressing sphalerite and pyrite. Therefore, a high-grade bulk concentrate should have enrichment of Pb and Cu while the grades of Fe and Zn should be minimized. In this work, in-house synthesized chitosan-polyacrylamide co-polymers (C-PAMs) of different structural characteristics were used as novel pyrite depressants, while sphalerite was depressed using zinc sulfate, which is usually used at industrial operations. The ML models were developed using nine different inputs (chitosan degree of deacetylation; weight ratio of chitosan: acrylamide; C-PAM dosage; sphalerite depressant dosage; collector dosage; frother dosage; slurry pH; flotation time; and impeller speed) with eight different outputs [grades and recoveries of lead (Pb), copper (Cu), iron (Fe), and zinc (Zn)]. Statistical performance metrics [coefficient of determination ($R^2$) and root mean square error (RMSE)] were used to evaluate the prediction performance of the ML models. Zeta-potential, total organic carbon (TOC), and X-ray photoelectron spectroscopy (XPS) measurements were used to investigate the adsorption mechanism of C-PAM on mineral surfaces.

## 2. Materials and Methods

### 2.1. Mineral Samples and Reagents

Model sulfide mineral samples of galena (PbS), chalcopyrite ($CuFeS_2$), and pyrite ($FeS_2$) were purchased from Ward's Science, USA. The mineral samples were crushed using a mortar and pestle, then dry-sieved. The $-38$ μm size fraction was used for reagent adsorption tests, including zeta-potential, contact angle, and XPS, while the $-75 + 38$ μm size fraction was used for TOC tests. According to the chemical analysis of the model sulfide minerals and X-ray diffraction results shown in Supplementary Information (Figure S1), the purities of all the model mineral samples were greater than 85%. The complex sulfide ore, containing PbS, $CuFeS_2$, and sphalerite (ZnS) as the valuable sulfide minerals and $FeS_2$ as the major sulfide gangue mineral, was obtained from a mine in North America. The flotation feed assayed 1.37% Pb, 0.3%Cu, 0.46% Zn, and 3.46% Fe. Sodium isopropyl xanthate (SIPX), zinc sulfate ($ZnSO_4$), and methyl isobutyl carbinol (MIBC) that were used as a collector, and a ZnS depressant and a frother, respectively, were all purchased from Fischer Scientific, Waltham, MA, USA. In-house synthesized C-PAM was used as a depressant of $FeS_2$. Sodium hydroxide (NaOH) and hydrochloric acid (HCl) were used to regulate slurry pH in the froth flotation tests and other adsorption tests. Deionized water was used in all the adsorption studies, while tap water was used in the batch flotation tests of the complex sulfide ore. Chitosan polymer (85% degree of deacetylation) and acrylamide monomer that were used in the synthesis of C-PAM were purchased from Fischer Scientific, USA. Cerium (IV) ammonium nitrate that was used as a polymerization initiator was purchased from Acros Organics. Acetic acid, acetone, and nitric acid used for the preparation of C-PAMs were all purchased from Fisher Scientific, Waltham, MA, USA.

### 2.2. Preparation of Chitosan-Polyacrylamide Copolymers (C-PAMs)

A total of nine C-PAM polymers (pyrite depressants) were synthesized in-house using chitosan polymer as the backbone and acrylamide monomer (AM) as side chains. The full procedure for the synthesis of C-PAM was adopted from the previous work of Monyake and Alagha [8,20]. The synthesized C-PAMs have variable chitosan degree of deacetylation and chitosan: AM weight ratio. The synthetic scheme of C-PAM is presented in Figure 1. Attenuated Total Reflectance Fourier Transform Infrared (ATR-FTIR) and Hydrogen Nuclear Magnetic Resonance ($^1$H-NMR) were used to characterize the synthesized C-PAMs, as

shown in Figures S2 and S3, respectively, in the Supplementary Information. For the NMR tests, C-PAM samples were dissolved in D4 acetic acid ($CD_3COOD$) and $D_2O$ solution, and chemical shifts were collected at 400 MHz. The FTIR signals and NMR chemical shifts of major functional groups of C-PAM are as follows:

NMR (ppm): 3.0 (H2 proton), 3.5 (H3–H6), 1.5 (methylene H), 2.06 (methine H), and 1.9 (methyl H).

FTIR ($cm^{-1}$): 3350 (OH), 3200 (NH stretching vibrations), 1650 (C=O), 1550 (N–H bending) [21].

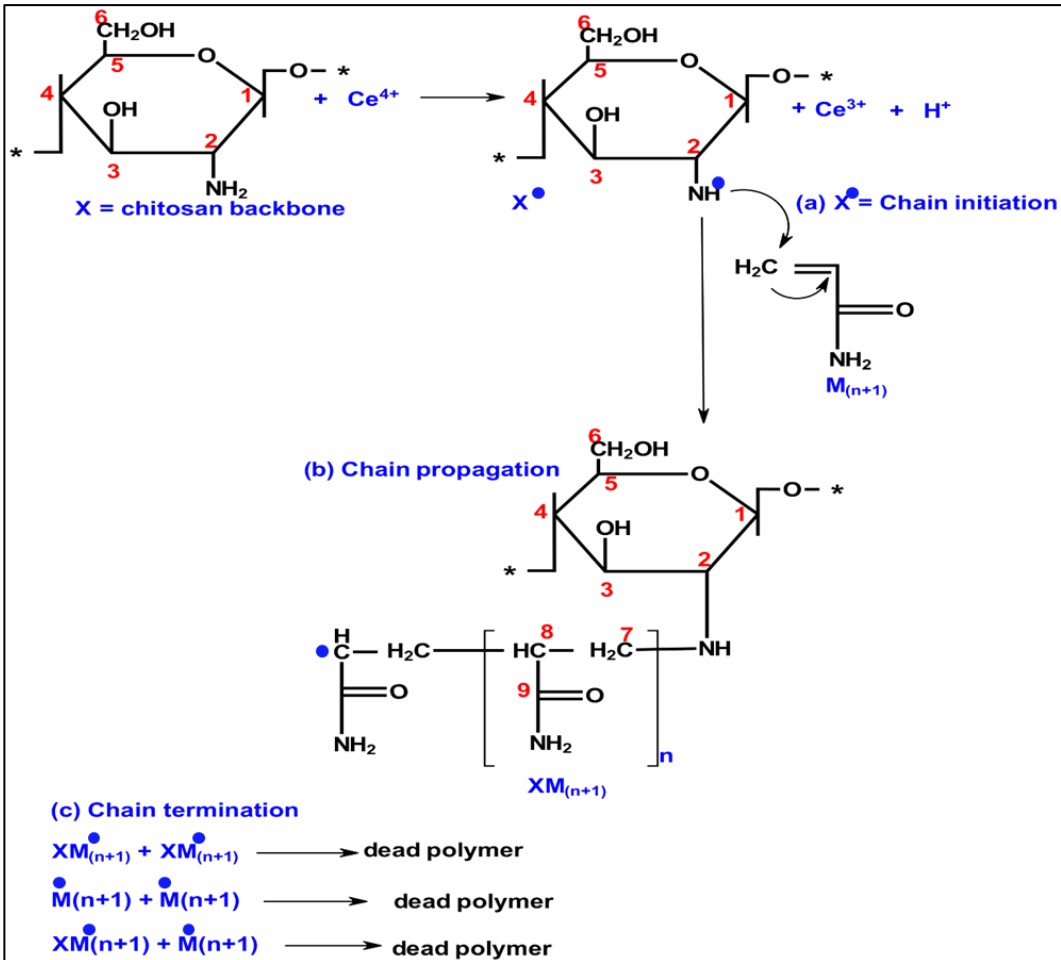

**Figure 1.** Synthetic scheme of chitosan-polyacrylamide copolymers (C-PAMs) used in this study (**a**) Chain initiation on the chitosan backbone; (**b**) Continuous chain propagation and (**c**) chain termination at the end of the reaction. Adopted from [21].

### 2.3. Froth Flotation Experiments

A total of 130 experiments (Tables S1–S4, Supplementary Information) were carried out wherein 9 flotation parameters were varied, as indicated in Table 1. Industrial procedures for this type of ore (MVT) initially involve a rougher flotation stage wherein galena and chalcopyrite are floated together as a bulk concentrate (rougher concentrate) using xanthate collectors while sphalerite and pyrite are depressed using zinc sulfate and sodium cyanide, respectively. The chalcopyrite/galena bulk concentrate is further processed in the cleaning stages (cleaner flotation) where chalcopyrite is floated, and galena is depressed. Sphalerite is usually floated from the rougher tailings in the following stages (cleaner flotation), while pyrite is depressed. This process is shown in the simplified flowsheet in Figure 2. Flotation tests in this work were performed in a single stage to represent the rougher flotation stage for this type of ore.

**Table 1.** Summary of the flotation variables (inputs) and flotation responses (outputs).

| Input Name | Input Code | Input Levels | | | Outputs |
|---|---|---|---|---|---|
| | | Low | Center | High | |
| Chitosan Degree of deacetylation (%) | $X_1$ | 75 | 85 | 95 | Fe grade (%) |
| Chitosan: AM Ratio (g/g) | $X_2$ | 3 | 5 | 7 | Fe recovery (%) |
| C-PAM dosage (g/ton) | $X_3$ | 75 | 100 | 125 | Pb grade (%) |
| Slurry pH (unitless) | $X_4$ | 7 | 8.5 | 10 | Pb recovery (%) |
| Xanthate Dosage (g/ton) | $X_5$ | 300 | 400 | 500 | Cu grade (%) |
| $ZnSO_4$ (g/ton) | $X_6$ | 500 | 600 | 700 | Cu recovery (%) |
| MIBC Dosage (g/ton) | $X_7$ | 50 | 75 | 100 | Zn grade (%) |
| Impeller Speed (RPM) | $X_8$ | 1000 | 1250 | 1500 | Zn recovery (%) |
| Flotation time (min) | $X_9$ | 3 | 5.5 | 8 | |

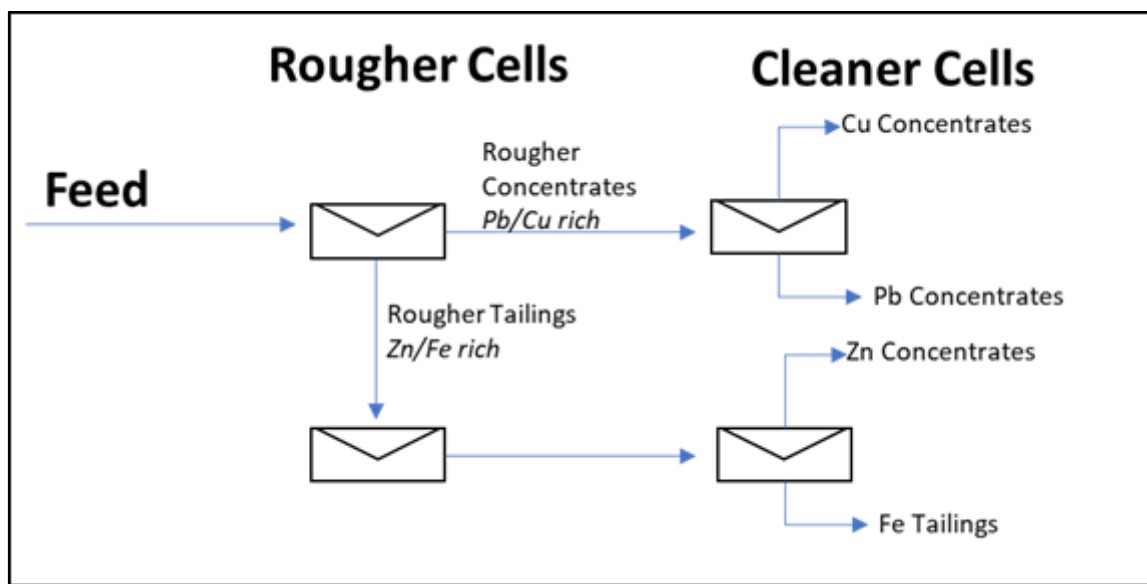

**Figure 2.** A simplified flow diagram of the typical flotation procedures of Mississippi Valley Type (MVT) complex sulfide ore was used in this study.

All flotation tests were carried out using a Denver flotation machine with a 1-L tank using tap water as a medium of flotation at 45 wt.% solids. In all flotation tests, the depressants were sequentially added first (C-PAM and $ZnSO_4$), followed by collector (SIPX) and frother (MIBC), which were each conditioned for 2–3 min. After flotation, the concentrates were collected, dried, and assayed for Pb, Cu, Zn, and Fe using atomic absorption spectroscopy (AAS), and metal recoveries in the concentrates were calculated using Equation (1).

$$Metal\ recovery = Cc/Ff * 100\% \tag{1}$$

where *C* and *F* represent the dried weights of the flotation concentrate and feed, respectively; whereas *c* and *f* are the metal grades in the concentrate and feed, respectively.

### 2.4. Adsorption Mechanism Studies

The following tests were carried out to understand the mechanism of pyrite depression by C-PAMs. Unless otherwise stated, freshly ground mineral particles of −38 μm were used in all adsorption tests.

### 2.4.1. Zeta Potential

Zeta potential tests were carried out using the Malvern Nano ZS zetasizer (Malvern instruments, Westborough, MA, USA) to study the interaction between model sulfide

minerals and C-PAM at a solid–liquid interface in potassium chloride (KCl) solution (0.001 M) as the background electrolyte. Each zeta potential test was performed at variable pH at a solid concentration of 0.01 wt.%. C-PAM (3 mg/L) was added separately and magnetically stirred for 3 min. After the minerals' suspensions settled, zeta potential measurements were performed using the supernatant. The reported zeta potentials were averaged from three independent zeta potential measurements.

2.4.2. Total Organic Carbon (TOC)

The adsorption density of C-PAM on galena, chalcopyrite, and pyrite were studied in the form of total organic carbon using the TOL-L series Total Organic Carbon analyzer-L Shimadzu Corporation, Columbia, MD, USA. Model mineral particles of size fraction $+38 - 75$ μm were used as they represent the intermediate particle size distribution usually used for micro flotation studies [20]. TOC was collected for pyrite before and after treatment with C-PAM at mineral suspension pH of 4, 7, and 10, while C-PAM dosage was varied at 25, 50, 75, 100, and 125 mg/L, respectively. Mineral suspensions in the presence of deionized water and a predetermined amount of C-PAM were mixed for 30 min. The supernatant was collected, filtered, and centrifuged to remove any invisibly suspended solids and used to measure TOC. The TOC value was regarded to represent the C-PAM concentration remaining in solution while the amount of C-PAM adsorbed on pyrite at equilibrium was calculated using Equation (2):

$$Q_e = \frac{(C_0 - C_e)V}{m} \tag{2}$$

where $Q_e$ is the amount of C-PAM adsorbed on pyrite surfaces at equilibrium (mg/g); $C_0$ and $C_e$ represent the initial and remaining concentration (mg/L) of C-PAM in the solution, respectively; $V$ is the solution volume (L); and $m$ represents the mass of each mineral sample (g).

2.4.3. X-ray Photoelectron Spectroscopy (XPS)

X-ray photoelectron spectroscopy (XPS) spectra were collected using the Kratos Axis 165 photoelectron spectrometer following the same procedure as TOC tests using $-38$ μm mineral particle size. After the mineral samples were completely settled, the solution was discarded, and the mineral samples were washed and filtered with deionized water and fully dried. The X-ray source used for XPS tests was monochromatic Al at 60 μm spot size. XPS spectra of untreated minerals and minerals treated with C-PAM were collected and analyzed using CasaXPS instrument software with Gaussian-Lorentzian peak resolving capabilities.

*2.5. Overview of the Machine Learning Models*
2.5.1. Random Forest Model

RF model is developed from the classification and regression tree (CART) model with the bagging technique [22,23]. The RF model grows hundreds of independent CARTs in a parallel manner. Each CART splits in a binary fashion at each node, and the entire CART grows in a recursive fashion. The CART stops growing when data in terminal nodes becomes near homogenous. The RF model allows CARTs to grow their maximum sizes without pruning and smoothing. The final output of the RF model is the average value of outputs from all trees. A major feature—two-stage randomization [24,25]—is distinct from the RF model with other CART-based models. The first randomization is that the bootstraps utilized to grow CARTs randomly select data from the parent dataset. The second randomization is that randomly selected variables are employed to determine the optimal split scenario at each node. The two-stage randomization ensures CARTs in the RF model are independent [26]. To achieve the optimal prediction, a 10-fold cross-validation (CV) method and grid search method are used to determine the optimal hyperparameters for the RF model. In this study, all predictions are unitized 100 trees in the forest and

4 splits at each node. The random forest classification model uses a predetermined number of independent trees organized into subsets of training data [22,27]. The number of trees is determined from the original dataset, where the optimum number of trees is approximately 66.66% of the training data set [22,27]. In this work, 100 trees were chosen to be almost 66.66% of the training dataset, and it should be noted that when 100 trees were exceeded, the random forest performance did not change. The optimum value of the number of nodes in this study was determined to be four to ensure low variance and decorrelation in the forest.

### 2.5.2. Artificial Neural Network (ANN)

Artificial neural network (ANN) is one of the most widely used computation techniques developed based-off on the complicated functioning of the human brain [28]. The basic structure of ANN consists of three different types of layers, i.e., input layer, output layer, and hidden layer [29]. Figure 3 displays the basic architecture of an artificial neural network with a single hidden layer and a single response variable. All of the processing in the neural network is carried out by these number of small yet powerful units known as neurons or nodes. A simple shallow neural network model consists of a single input layer with multiple input nodes, a single output layer with one node, and a single hidden layer with multiple neurons optimized for any given problem. The neurons are connected with both the input and output layers, with every neuron forming a unique connection with each of the neighboring layer nodes. The connections between these adjacent hidden layer neurons or between neurons and the nodes are distinguished based on the assigned mathematical weights, which are later optimized. These weights govern the influence each neuron holds over the output of the previous layer that gets transferred to its immediately connecting neurons in the subsequent layer [30]. The number of neurons in both the input and output layers depends upon the quantity of input and output variables used for model development, respectively. The number of neurons in a hidden layer is considered a hyperparameter for neural network architecture and thus requires optimization during model training.

The output of any neuron in the hidden layer(s) or in the output layer is mathematically computed using Equation (3) [31–36]. A bias term is added to the computed output to ensure a non-zero result by a neuron, even when zero inputs are given. The output generated by every neuron is taken through an activation function to ensure advanced non-linear learning by the neural network [37].

$$z_k = f_0 \left[ \sum_{j=1}^{H_N} W_{kj} \times f_h \left( \sum_{i=1}^{I_N} W_{ji} \times X_i + W_{j0} \right) + W_{k0} \right] \quad (3)$$

where,

$W_{ji}$: Connection weight connecting the neurons of $j$th hidden layer with the node of $i$th input layer;
$W_{j0}$: Bias added to the neuron of $j$th hidden layer;
$f_h$: Activation and transfer function for each hidden neuron;
$W_{kj}$: Connection weight connecting the node of $k$th output layer with the neuron of $j$th hidden layer;
$W_{k0}$: Bias added to the node of output layer;
$f_0$: Activation and transfer function for output node;
$X_i$: $i$th input variable;
$Z_k$: $k$th response variable;
$I_N$: Total input layer nodes in the model;
$H_N$: Total hidden layer neurons in the model.

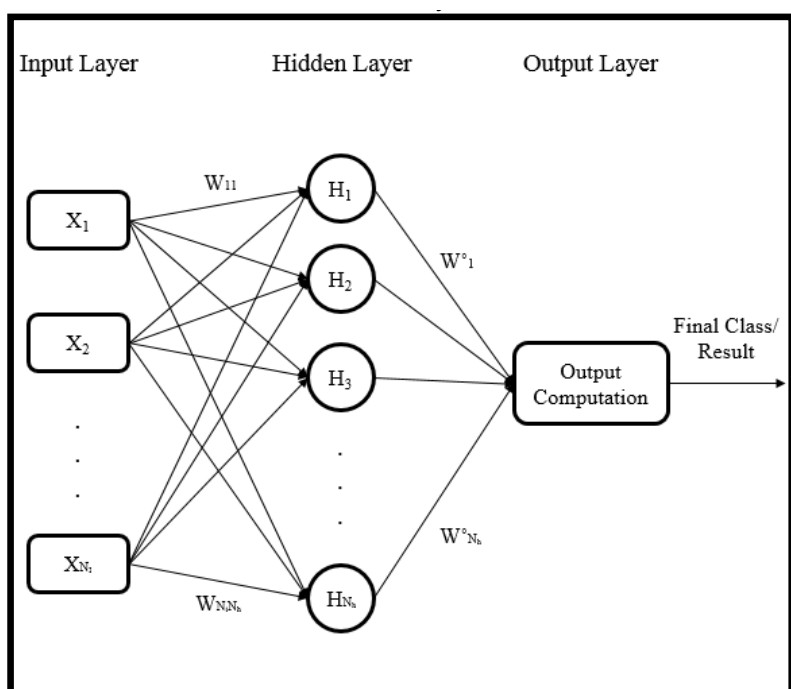

**Figure 3.** Basic representation of single hidden layer feed artificial neural network with single output node [38].

### 2.6. Data Collection and Dataset Preparation

Experimental datasets were derived from a total of 130 laboratory batch flotation experiments (rougher flotation). Nine different flotation variables (inputs) were tested at three different levels each while the metal grades and recoveries were regarded as the outputs. The significance of various input variables for the prediction of grades and recoveries of Cu, Pb, and Fe is shown in Figures S4 and S5 in the Supplementary Information. The data were cleaned to detect and remove any errors. Statistical parameters for the dataset consisting of 130 data-records featuring the nine input variables and the 8 corresponding outputs have been given in Table 2. Before the data was used for model training, scaling was done through employing linear mapping, as given by Equation (4), because each of the input variables were widely distributed. Scaled data was then used for model development.

$$X\_n\,[0, 1] = (X - X\_min)/(X\_max - X\_min) \tag{4}$$

where,

$X\_n$ = Scaled/Normalized value for input variable '$x$';
$X\_max$ = Highest value of input variable '$x$';
$X\_min$ = Minimum value of input variable '$x$';
$Z$ = Real value of input variable '$x$'.

**Table 2.** Statistical parameters of the nine inputs and eight outputs make up the ML database.

| Attribute | Minimum | Maximum | Mean | Standard Deviation |
|---|---|---|---|---|
| Chitosan Degree of deacetylation (%) | 75.00 | 95.00 | 85.00 | 5.55 |
| Chitosan: AM Ratio (g/g) | 3.00 | 7.00 | 5.00 | 1.11 |
| C-PAM dosage (g/ton) | 75.00 | 125.00 | 100.00 | 13.87 |
| Slurry pH (unitless) | 7.00 | 10.00 | 8.50 | 0.83 |
| Xanthate Dosage (g/ton) | 300.00 | 500.00 | 400.00 | 55.47 |
| $ZnSO_4$ (g/ton) | 500.00 | 700.00 | 600.00 | 55.47 |
| MIBC Dosage (g/ton) | 50.00 | 100.00 | 75.00 | 13.87 |

**Table 2.** *Cont.*

| Attribute | Minimum | Maximum | Mean | Standard Deviation |
|---|---|---|---|---|
| Impeller Speed (RPM) | 1000.00 | 1500.00 | 1250.00 | 138.68 |
| Flotation time (min) | 3.00 | 8.00 | 5.50 | 1.39 |
| Pb grade (%) | 7.29 | 24.39 | 15.76 | 4.00 |
| Pb recovery (%) | 19.87 | 99.94 | 64.77 | 22.68 |
| Fe grade (%) | 0.65 | 11.36 | 7.63 | 1.91 |
| Fe recovery (%) | 2.43 | 64.97 | 35.00 | 11.21 |
| Cu grade (%) | 1.14 | 2.81 | 1.93 | 0.32 |
| Cu recovery (%) | 23.74 | 85.75 | 55.97 | 16.45 |
| Zn grade (%) | 0.50 | 7.22 | 2.31 | 1.78 |
| Zn recovery (%) | 1.09 | 22.82 | 8.45 | 5.05 |

## 3. Results

### 3.1. Adsorption Mechanism of C-PAM on Mineral Surfaces

### 3.1.1. Zeta Potential

These tests specify the strength of C-PAM adsorption on mineral surfaces. Strong adsorption of C-PAM was indicated by a large change in the zeta potential of mineral suspensions, while a weak adsorption was revealed through a smaller shift in the zeta potential value. Figure 4 shows the effects of pH on the zeta potential of aqueous suspensions of model sulfide minerals in the absence and presence of C-PAM. As shown in Figure 4a, pyrite exhibited a positive charge at a pH lower than 6.5 and reached a point of zero charge at pH 6.5, also referred to as the isoelectric point (IEP). The surfaces of pyrite were observed to be negatively charged at mildly alkaline to strongly alkaline pH. It is common for sulfide minerals to be negatively charged at a pH exceeding 2 because they contain sulfur ions which have an IEP of 1.6, together with negatively charged metal hydroxide species [39]. In the absence of C-PAM, galena was positively charged and reached the IEP at pH 5 and became negatively charged at pH > 5, as shown in Figure 4b. Chalcopyrite IEP was expected to be around pH 2–3 [40], but as observed in this work, chalcopyrite was negatively charged at all pH ranges tested, as shown in Figure 4c, without the addition of C-PAM. When C-PAM was introduced as a depressant, all minerals exhibited more positive shifts of zeta potential values ($\Delta\zeta$), as shown in Figure 4a–c. The zeta potential of C-PAM used was +48.98 mV; therefore, it was expected that a positively charged (cationic) depressant would electrostatically adsorb on model mineral surfaces that were negatively charged at an alkaline pH range.

C-PAM was observed to strongly adsorb on pyrite surfaces at pH 8–12, as shown by the large positive shift in zeta potentials as compared to other sulfide minerals. Table 3 shows the anticipated strength of C-PAM adsorption on metal sulfides as indicated by the magnitude of $\Delta\zeta$ of mineral suspensions after C-PAM adsorption. It was observed that at pH 8, 10, and 12, $\Delta\zeta$ values for pyrite were +28.6, +34.4, and +35.5 mV, respectively. Even though positive shifts of the zeta potential values of galena and chalcopyrite were observed, however, $\Delta\zeta$ values were observed to be smaller as compared to those of pyrite. For example, at pH 8, 10, and 12, $\Delta\zeta$ values were +9.3, +17.3, and +24.8 mV, respectively, for galena, and +22.2, +20.5, and +20.2 mV, respectively, for chalcopyrite. These results indicated that C-PAM is anticipated to have preferential adsorption on pyrite minerals at alkaline pH. As published literature, the depression of pyrite minerals increases with increasing pH [41–44]; therefore, this was a good indicator that C-PAM might potentially be a desirable depressant of pyrite in an alkaline environment.

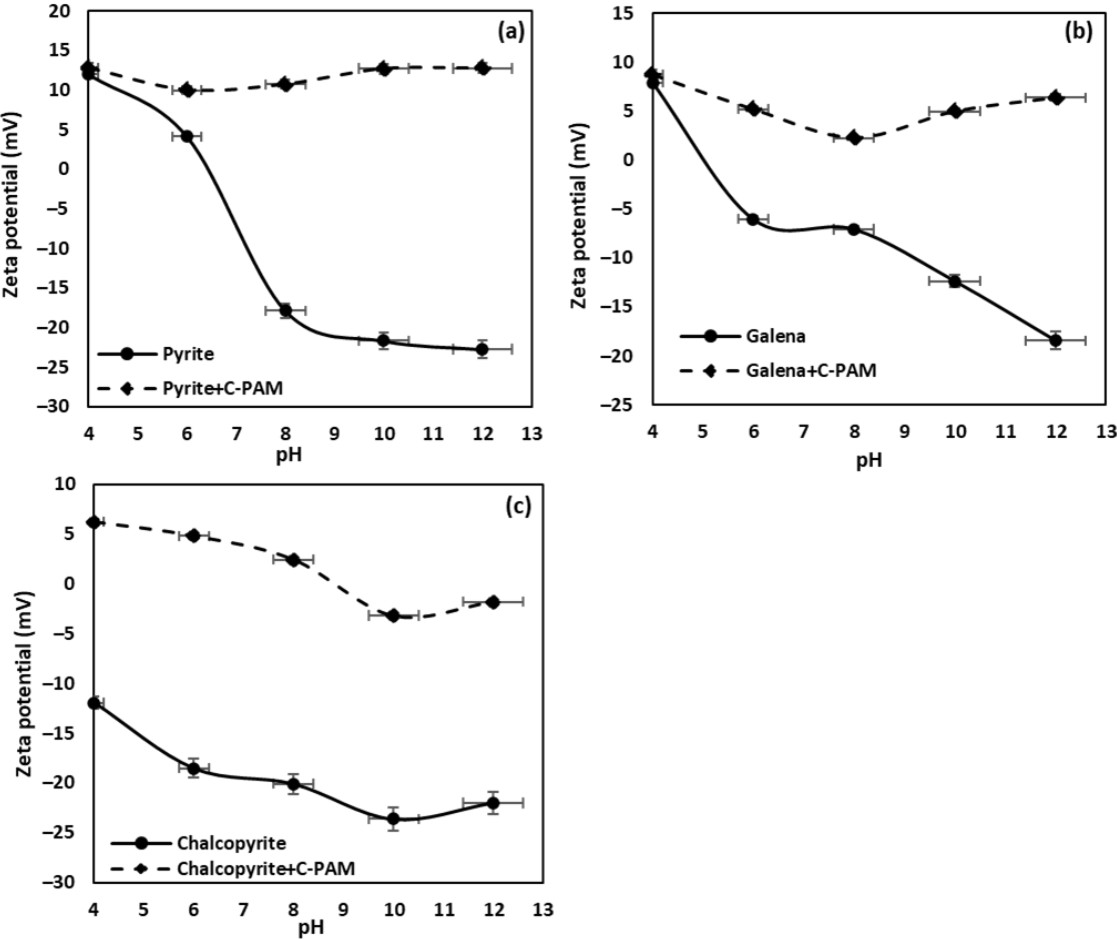

**Figure 4.** Zeta potential measurements of model sulfide minerals before and after C-PAM addition (**a**) pyrite, (**b**) galena, and (**c**) chalcopyrite.

**Table 3.** Magnitude of shifts of zeta potential of metal sulfides' suspensions ($\Delta\zeta$) post C-PAM adsorption at different pH.

| pH | Zeta Potential Shifts ($\Delta\zeta$), mV | | |
|---|---|---|---|
| | Galena | Chalcopyrite | Pyrite |
| 4 | +0.8 | +18.2 | +0.8 |
| 6 | +11.2 | +23.3 | +5.9 |
| 8 | +9.3 | +22.5 | +28.6 |
| 10 | +17.3 | +20.5 | +34.4 |
| 12 | +24.8 | +20.2 | +35.5 |

3.1.2. Adsorption Density Analysis

The adsorption behavior of C-PAM on the surfaces of model pyrite, galena, and chalcopyrite minerals is shown in Figure 5 in the form of the amount of C-PAM adsorbed on mineral surfaces at equilibrium as a function of initial C-PAM dosage in the mineral suspensions at pH 7. For all minerals, the amount of C-PAM adsorbed increased with the increase in the initial concentration of C-PAM in the mineral suspensions. It is clear from Figure 5 that the adsorption of C-PAM was stronger on pyrite surfaces as compared to galena and chalcopyrite at an initial concentration range between 75–125 mg/L. This suggested that this concentration range of C-PAM could be ideal for the depression of pyrite in the bulk flotation of chalcopyrite and galena.

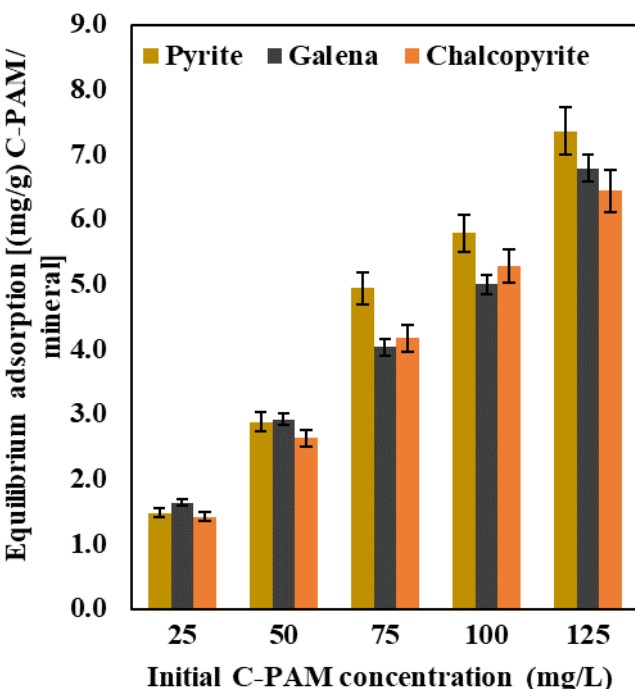

**Figure 5.** Adsorption densities of C-PAM on model sulfide minerals as a function of initial C- PAM concentration measured using total organic carbon (TOC) analysis.

An adsorption isotherm was established for the adsorption of C-PAM on model sulfide minerals which followed Langmuir adsorption isotherm, indicating monolayer adsorption shown in Figure 6. At the tested concentration range of C-PAM (25–125 mg/L), the maximum adsorption on pyrite was found to be 35.21 mg/g while maximum adsorption on chalcopyrite and galena were 22.42 mg/g and 12.45 mg/g, respectively. These results suggest that C-PAM had preferential adsorption on pyrite compared to galena and chalcopyrite, which is consistent with the work of Monyake and Alagha [20] on model mineral samples when the tested range of C-PAM concentrations were lower (1–5 mg/L). These findings could be crucial to the flotation of a complex sulfide ore where pyrite is selectively depressed in the first stage of flotation while galena and chalcopyrite are floated in bulk.

### 3.1.3. X-ray Photoelectron Spectroscopy (XPS) Analysis

Zeta potential and adsorption density studies have suggested that C-PAM preferentially adsorbed on pyrite surfaces as compared to galena and chalcopyrite, which could enhance the bulk flotation of galena and chalcopyrite. As a result, X-ray photoelectron spectroscopy (XPS) measurements were used to delineate the adsorption mechanism of C-PAM on pyrite surfaces. The distinctive binding energies of electrons of each element identified by XPS have been used to identify chemical states and chemical composition of different samples [45], which is helpful when studying the adsorption mechanism of reagents on mineral surfaces [8,20]. Figure 7 shows the full range XPS spectra of pyrite surfaces before and after C-PAM treatment. As indicated in Figure 7a, pyrite exhibited the expected surface species of Fe 2p, S 2p, C 1s, and O 1s which was consistent with published reports [46]. The Fe 2p peaks were observed at 712.08 eV, while the S 2p peaks appeared at 162.08 eV. The presence of C 1s on pyrite at 284.58 eV was a result of adventitious carbon, while O 1s at 532.08 eV was due to possible mineral oxidation [46,47]. The concentrations of Fe, S, C, and O on the surface of pyrite were 7.08, 20.50, 41.76, and 30.69 at.%, respectively. However, after the pyrite was exposed to C-PAM, nitrogen species were observed on pyrite surfaces as shown by the new N 1s (1.53 at.%) peak at 399.68 eV in Figure 7b of pyrite treated with C-PAM. The concentrations of Fe and S decreased to 5.47 and 15.68 at.%, respectfully, after the addition of C-PAM, which supported that adsorption took place. On

the other hand, the concentrations of C and O species increased to 44.61 and 32.7 at.%, respectively, which suggested that C-PAM adsorbed on pyrite surfaces.

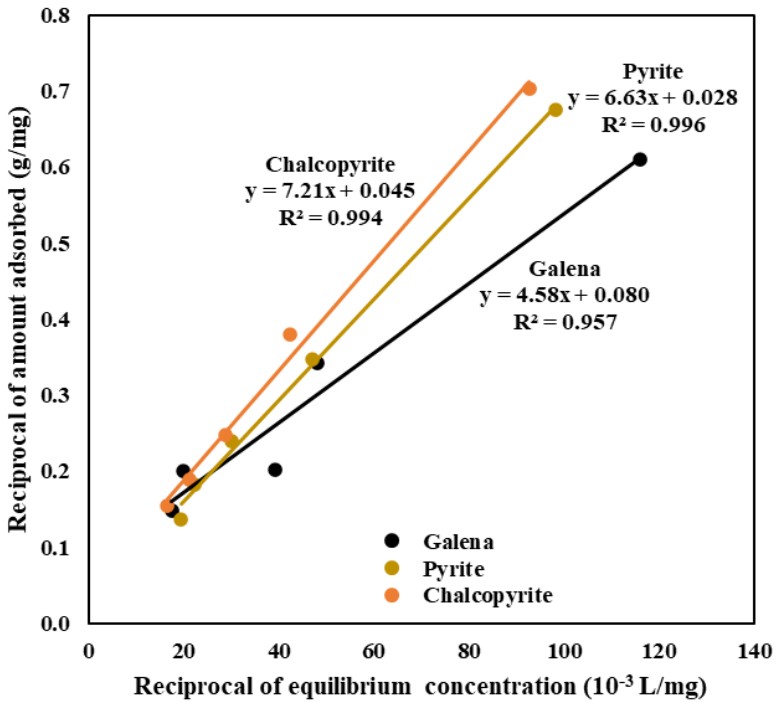

**Figure 6.** Langmuir adsorption isotherm model explaining the adsorption behavior of C-PAM on metal sulfides.

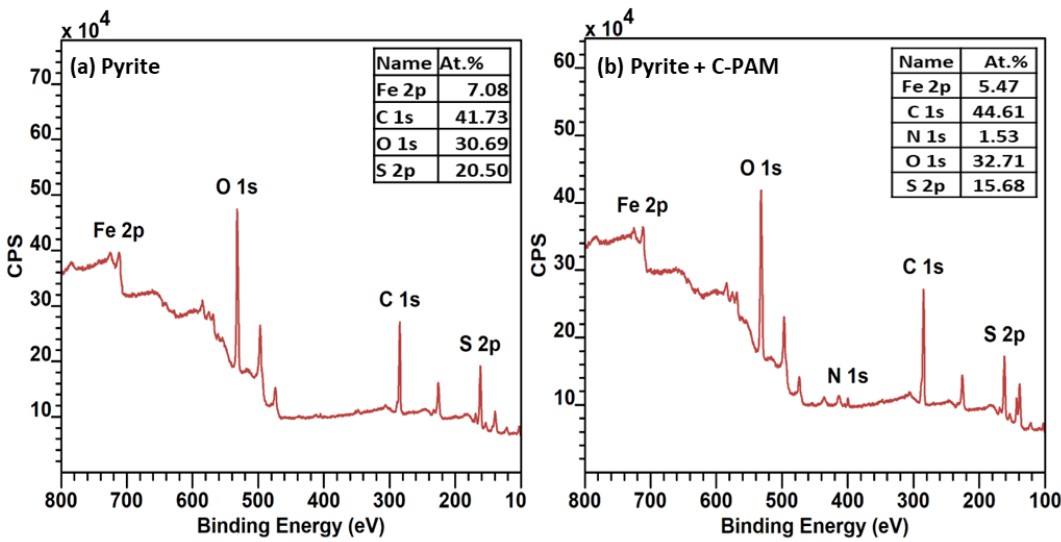

**Figure 7.** X-ray photoelectron spectroscopy (XPS) spectra of pyrite (**a**) before and (**b**) after C-PAM adsorption.

### 3.1.4. Mechanism of C-PAM Adsorption on Pyrite

Further assessment of the mechanism of C-PAM adsorption on pyrite was done through XPS. Functional groups of C-PAM accountable for the depression of pyrite were inspected through high-resolution spectra. Figure 8a,b shows the N 1s species of pyrite before and after C-PAM adsorption. The major peak at 399.68 eV was assigned to the nitrogen atoms in the primary amine group (–NH$_2$), while the second peak observed at 400.89 eV was originating from the O=C–NH$_2$ group. After C-PAM adsorption, the amine

group shifted by +0.11 eV while the amide group shifted by +0.62 eV with respect to the original N 1s peaks of C-PAM. It was reported that any binding energy shift of greater than 0.1 eV represented a chemical interaction between organic depressants and sulfide minerals when the interaction between corn starch and pyrite was studied [48]. As a result, chemisorption was proposed as the main adsorption mechanism of C-PAM on pyrite, as indicated by the changes in binding energy.

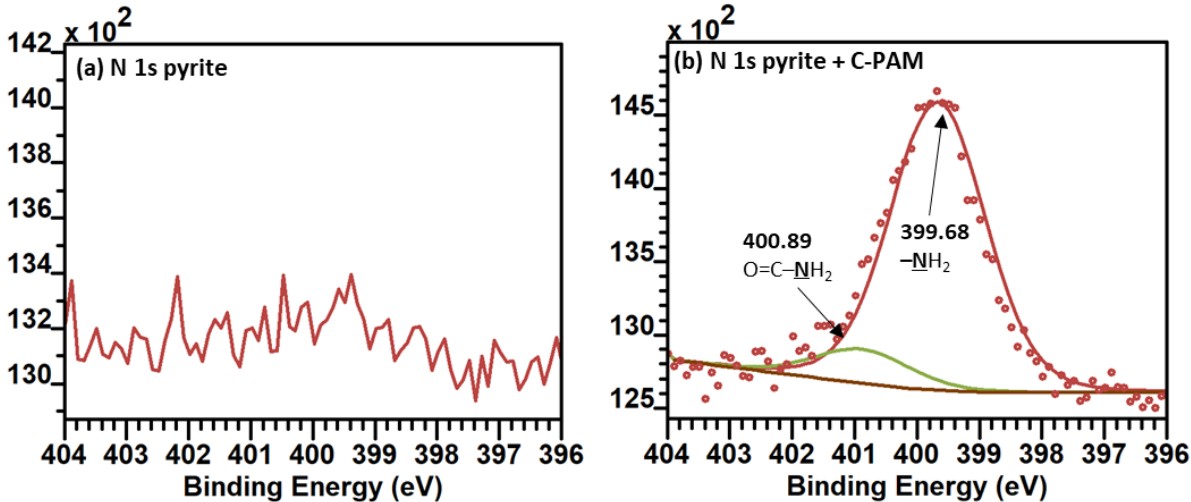

**Figure 8.** X-ray photoelectron spectroscopy (XPS) high-resolution N 1s spectra of pyrite (**a**) before and (**b**) after C-PAM adsorption.

Additionally, the C 1s spectra for the surface of pyrite before and after C-PAM treatment were obtained, as shown in Figure 9a,b. The peak at 284.42 eV was assigned to the C-C resulting from adventitious carbon [45,49] contamination on the surfaces of pyrite shown in Figure 9a. After treating pyrite with C-PAM, two new peaks were observed at 286.14 eV and 288.46 eV resulting from the C–OH and O=C–NH$_2$ of C-PAM, respectively, as shown in Figure 9b [45].

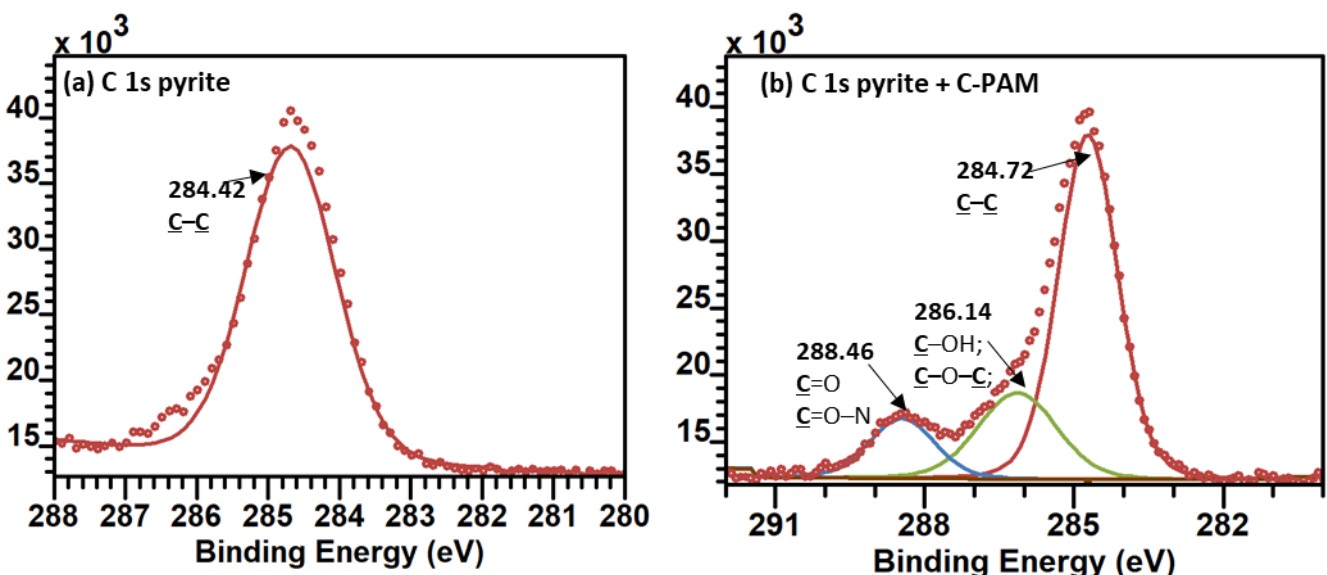

**Figure 9.** X-ray photoelectron spectroscopy (XPS) high-resolution C1s spectra of pyrite (**a**) before and (**b**) after C-PAM treatment.

### 3.2. Machine Learning Models Comparison

The type of ore tested in this work was the Mississippi Valley type (MVT) ore consisting of galena, chalcopyrite, and sphalerite as the main valuable sulfides while pyrite was the main gangue sulfide mineral. For this type of ore, the common procedure is to float galena and chalcopyrite as a bulk concentrate while depressing sphalerite and pyrite [20]. The performance of the RF model was compared to the ANN model in predicting the grades and recoveries of metals (Pb, Cu, and Fe) in the flotation concentrates.

### 3.2.1. Random Forest (RF) Model

The RF model was trained by the training dataset consisting of 75% of data records from the parent database. Then, the training dataset (containing the remaining data-records) was employed to assess the performance of the RF model on predictions of grades and recoveries of Pb, Cu, and Fe in the flotation concentrates. Predictions for the training and testing dataset as produced by the RF model are shown in Figures 10–12, and the statistical parameters pertaining to the RF model's performance are shown in Table 4. As shown in Figures 10–12 and Table 4, predictions, as produced from the RF model, of grades and recoveries of Pb, Cu, and Fe exhibit reasonable accuracy. $R^2$ pertaining to the prediction performance of Fe grade, Fe recovery, Pb grade, Pb recovery, Cu grade, and Cu recovery is 0.87, 0.91, 0.78, 0.63, 0.80, and 0.73, respectively. RMSE pertaining to the prediction performance of Fe grade, Fe recovery, Pb grade, Pb recovery, Cu grade, and Cu recovery is 0.834%, 4.196%, 1.962%, 12.402%, 0.141%, and 8.268%, respectively. It is expected that the RF model can produce accurate predictions because the RF model has a unique architecture that other models do not obtain [23,50,51]. First, due to two-stage randomization [23,25], all trees decorrelate with each other. This ensures that bias error is minimized. Next, the RF model constructure hundreds of independent trees without any smoothing and pruning. The variation errors for outputs are diminished due to the convergence of outputs from a large number of trees. Finally, all hyperparameters applied in the model are optimized by 10-fold cross-validation method [27] and grid-search method [25,52]. Therefore, the RF model finds the optimal underlying structures between inputs and output.

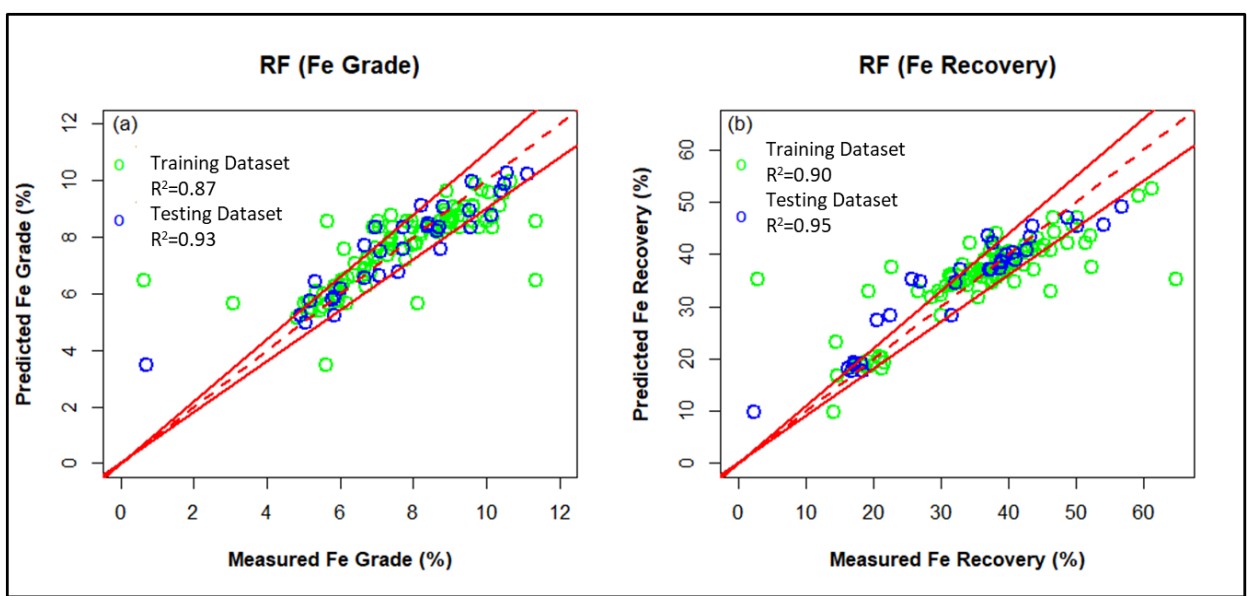

**Figure 10.** Measured vs. predicted (**a**) grades and (**b**) recoveries of iron (Fe) using the RF model for training and testing phases.

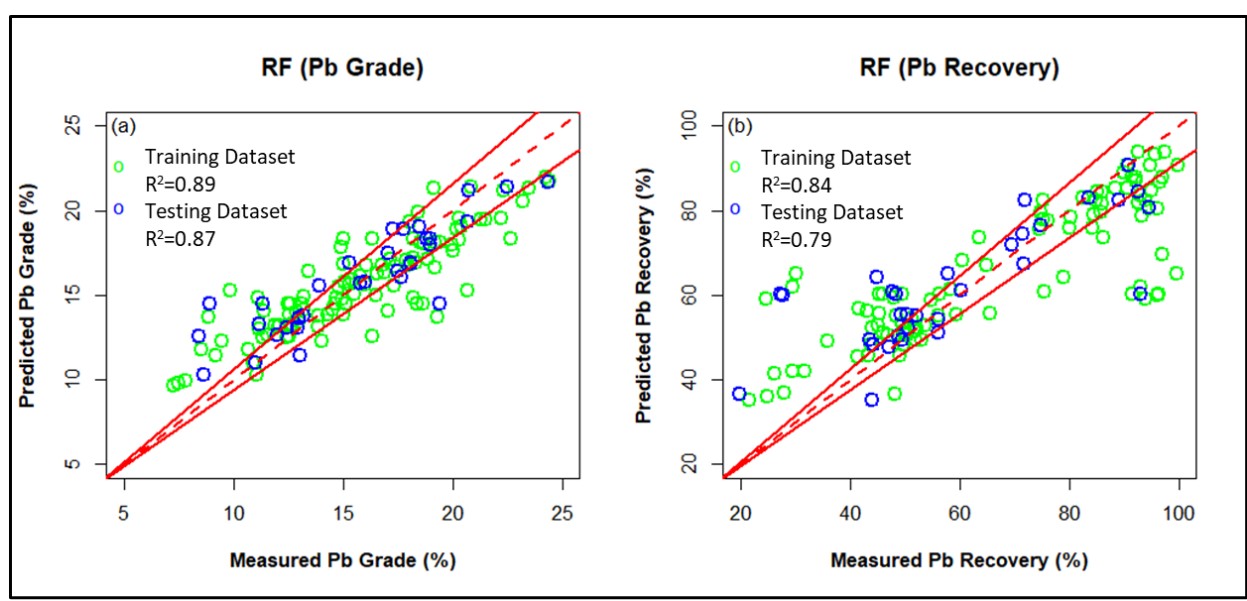

**Figure 11.** Measured vs. predicted (**a**) grades and (**b**) recoveries of lead (Pb) using the RF model for training and testing phases.

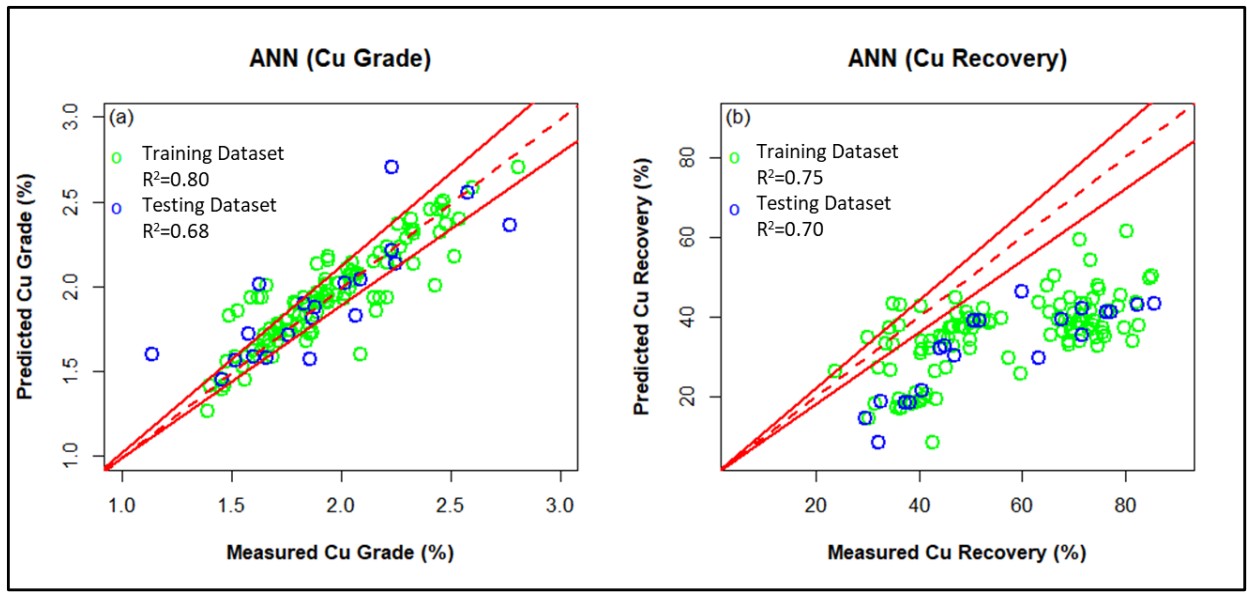

**Figure 12.** Measured vs. predicted (**a**) grades and (**b**) recoveries of copper (Cu) using the RF model for training and testing phases.

**Table 4.** Overall performance indicators for the developed ML models.

| Model | Training | | Testing | |
|---|---|---|---|---|
| | $R^2$ | RMSE | $R^2$ | RMSE |
| RF | 0.883 | 4.380 | 0.901 | 3.780 |
| ANN | 0.789 | 3.868 | 0.717 | 4.349 |

### 3.2.2. Artificial Neural Network (ANN) Model

The training dataset was used to develop a single hidden layer feed-forward neural network model with eight input variables and eight response variables. Two different neural network models were developed, one for the metal recoveries and the other one

for the output metal recoveries. The optimization algorithm, resilient backpropagation with weight backtracking, was used with the sum of square loss as the cost function during model training and error back propagation. The tangent hyperbolic function was used as the activation function at the hidden layer neuron connection, whereas the identity function was used as the activation function for the output layer, for both the neural network models. A 5-fold cross-validation was used to avoid the overfitting with an error threshold value of "0.05" during the model development process.

A search was conducted to optimize the hyperparameter for the neural network model through trial and error. The goal was to find the optimum number of hidden neurons in an attempt to optimize the single hidden layer neural network architecture [31,38,53]. The optimization results showed that the performance of the neural network improved as the number of neurons in the single hidden layer increased, then the performance of the model stabilized as the number of neurons reached a total of 41 and 53 for metal grades and metal recovery outputs, respectively. Increasing the neurons beyond these optimum numbers in the respective hidden layer did not improve the neural network performance. In fact, it would just result in increased computational cost. The two final neural network models were developed using the same training dataset with an optimum number of hidden neurons, 41 and 53, for metal grades and metal recovery outputs, respectively. The model development process was stopped as soon as convergence was reached for both neural network models. The two final optimized artificial neural network models are shown in Figures 13 and 14, consisting of optimum single hidden layer neural networks for metal grades and metal recovery outputs, respectively. Blue lines display the added bias terms for generating a non-zero output for the neuron/node, whereas black lines display the connections between the hidden neurons and nodes. Moreover, the thickness of the line connection between these neurons and nodes describes the relative importance in terms of the magnitude of the connection weights: the thicker the line, the higher the weight between the respective neuron and node. Once the final trained neural network models were developed, validation was carried out by using the test dataset to evaluate both the model's ability to predict the metal grade and recovery outputs for the flotation of base metal sulfides in the presence of C-PAM as a pyrite depressant.

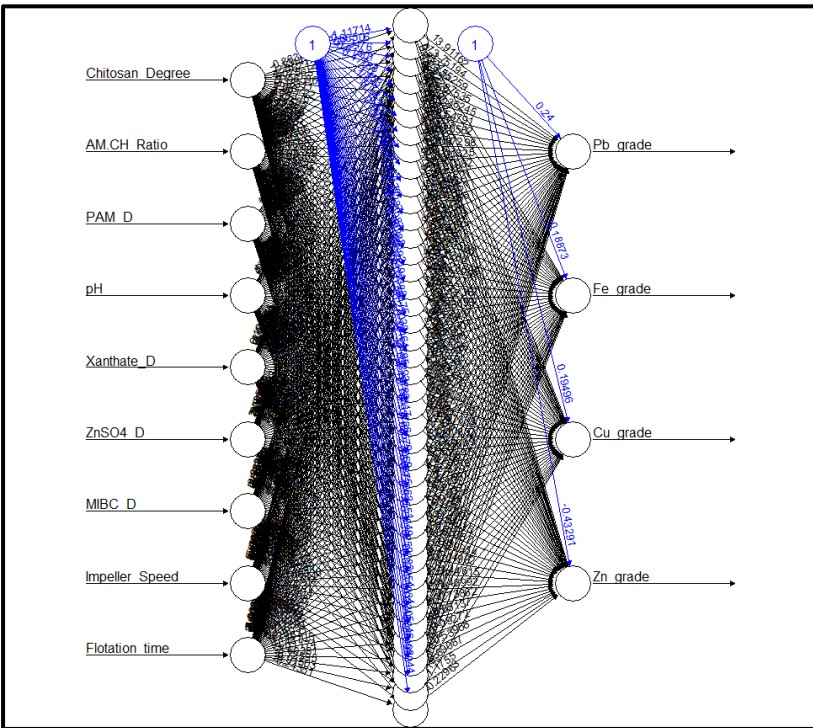

**Figure 13.** Final optimum artificial neural model for metal grade outputs.

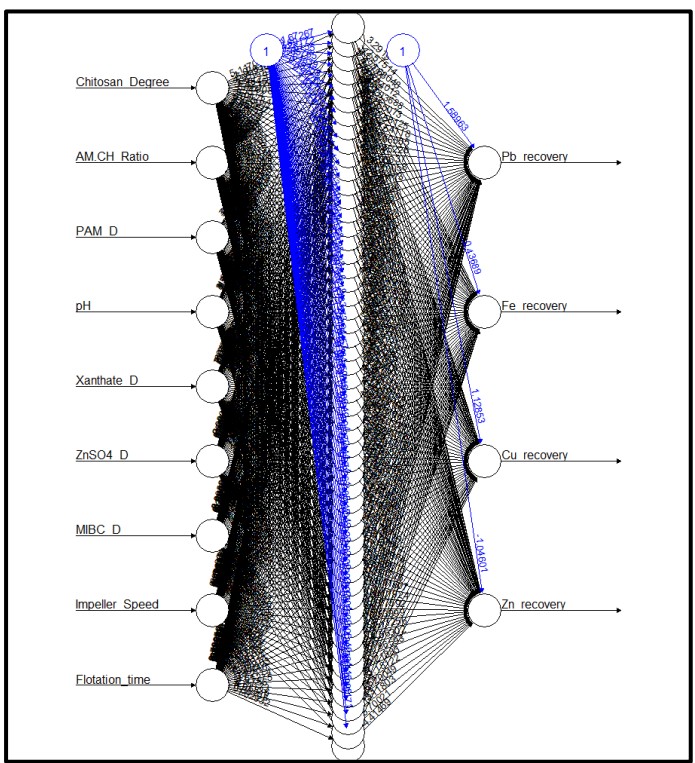

**Figure 14.** Final optimum artificial neural model for metal recovery outputs.

Figures 15–17 display the real vs. predicted plots, for both grade and recovery of iron, lead, and copper, as a graphical representation of the performance evaluation for both the neural network models. The plots show the results for both the training and testing phases of the neural network models. A 1:1 correlation line in each of those plots shows the goodness of fit between the actual metal and recovery values and the predicted corresponding output values by the respective models during both the training and testing stages.

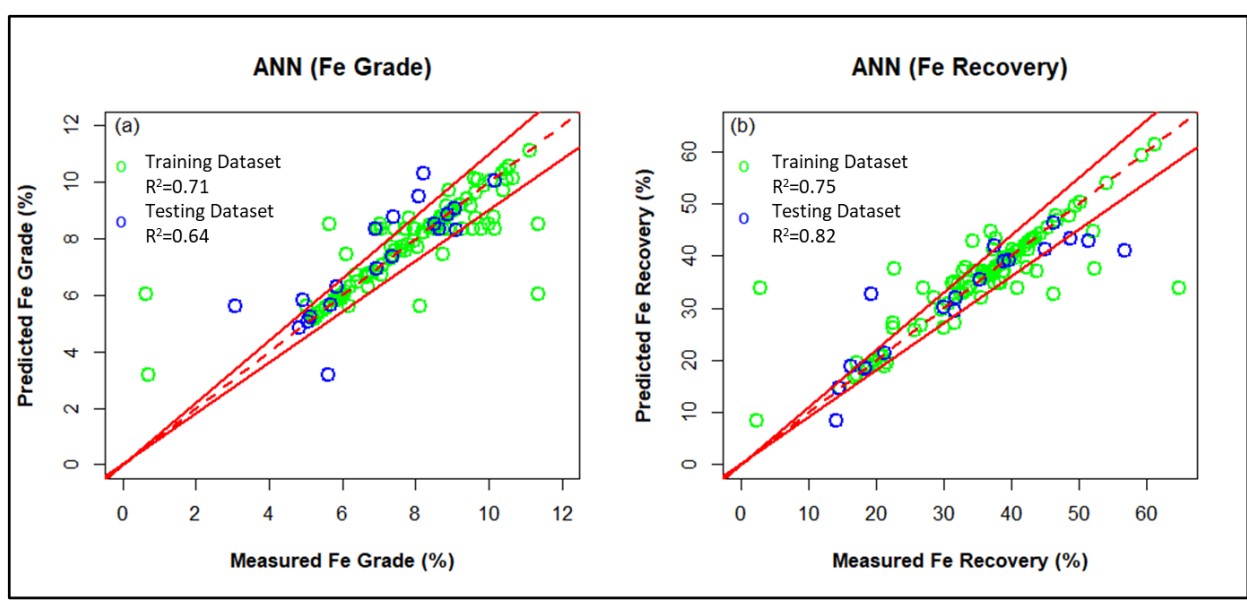

**Figure 15.** Measured vs. predicted (**a**) grades and (**b**) recoveries of iron (Fe) using the ANN model for training and testing phases.

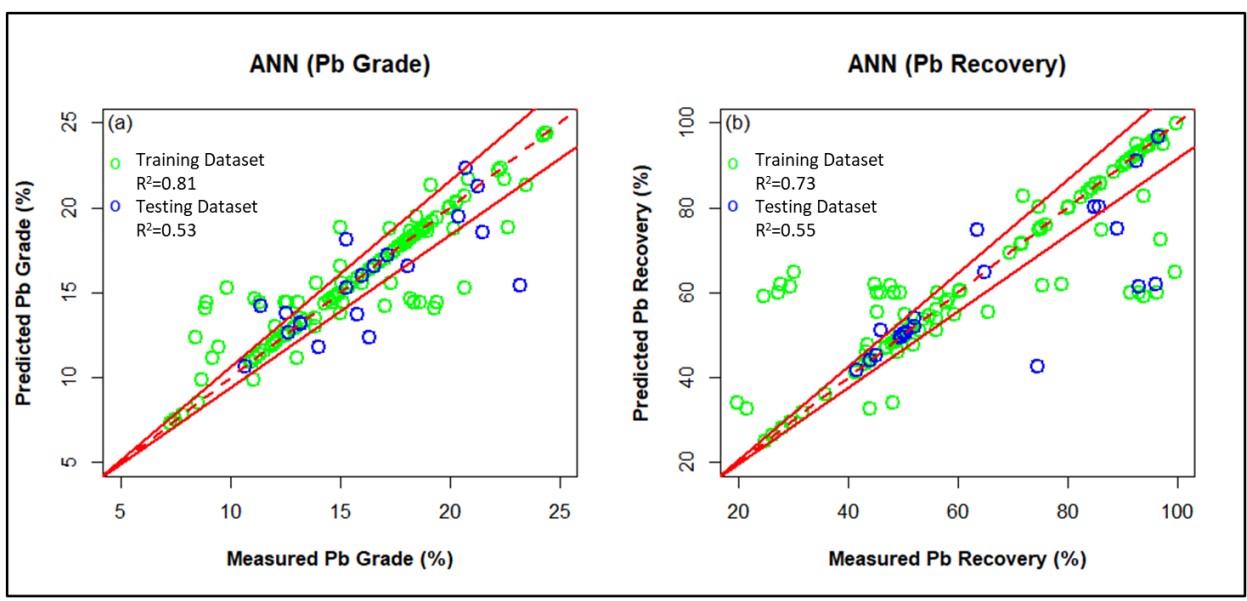

**Figure 16.** Measured vs. predicted (**a**) grades and (**b**) recoveries of lead (Pb) using the ANN model for training and testing phases.

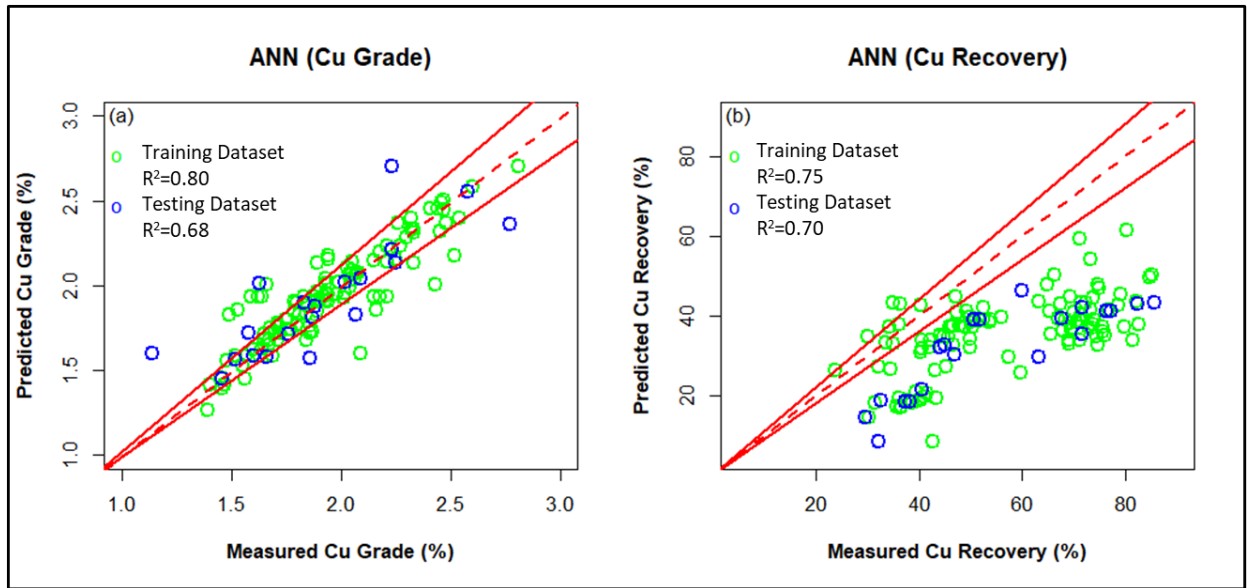

**Figure 17.** Measured vs. predicted (**a**) grades and (**b**) recoveries of copper (Cu) using the ANN model for training and testing phases.

### 3.3. Discussion on the Models' Prediction Performance

The data obtained from flotation experiments were analyzed and randomly divided into training (75%) and testing (25%) sets using the cross-validation method. Previous studies have successfully used the 75–25% data partition method for training and testing the dataset [19,26,54]. Each of the outputs was trained and tested separately. The data split was randomized but done in such a way that the training data set was representative of the parent dataset. Two statistical parameters were used to evaluate the prediction performance of the models. These parameters basically estimate the cumulative error in predictions of the metal grades and recoveries in the test dataset with respect to the experimental values [26,55]. The performance of the developed models was evaluated using the coefficient of determination ($R^2$) as shown in Equation (5) and root mean square error (RMSE) shown in Equation (6) [17]. These methods are the most commonly used statistical

indicators for evaluating the performance of models [56]. $R^2$ values range between 0 and 1, with values closer to 1 indicating that there is a high correlation. As for RMSE, a higher value indicates that there is a higher deviation of the predicted values from actual values, while a RMSE value closer to 0 indicates that there is a perfect match between the actual and predicted values [17]. Table 4 shows the overall performance metric evaluation results for both the machine learning algorithms, RF and ANN, whereas Table 5 displays a more detailed evaluation by breaking down the performance into respective grades and recoveries of Fe, Pb, and Cu. The ANN model for metal grade showed an $R^2$ value of 0.71, 0.81, and 0.80 for Fe, Pb, and Cu grade, respectively, during model training, and an $R^2$ value of 0.64, 0.53, and 0.68 for respective metal grades, during model testing, thus giving an overall $R^2$ value of 0.758 and 0.652, for training and testing phases, respectively. The RMSE values by the ANN model for metal grades came out to be 1.02, 1.77, and 0.14 for Fe, Pb, and Cu grades, respectively, during model training, whereas 1.11, 2.43, and 0.22 for respective metal grades, during model testing, thus giving an overall RMSE value of 4.75 and 5.42, for training and testing phases, respectively.

**Table 5.** Statistical performance indicators pertaining to the flotation responses (metal grades and recoveries).

| Response | ML Model | Training | | Testing | |
|---|---|---|---|---|---|
| | | $R^2$ (Unitless) | RMSE (%) | $R^2$ (Unitless) | RMSE (%) |
| Pb grade | RF | 0.75 | 2.077 | 0.79 | 1.962 |
| | ANN | 0.81 | 1.77 | 0.53 | 2.43 |
| Pb recovery | RF | 0.70 | 13.288 | 0.63 | 12.402 |
| | ANN | 0.73 | 12.02 | 0.55 | 13.44 |
| Cu grade | RF | 0.77 | 0.164 | 0.80 | 0.141 |
| | ANN | 0.80 | 0.14 | 0.68 | 0.22 |
| Cu recovery | RF | 0.70 | 9.298 | 0.73 | 8.268 |
| | ANN | 0.75 | 8.14 | 0.69 | 9.77 |
| Fe grade | RF | 0.61 | 1.143 | 0.87 | 0.834 |
| | ANN | 0.71 | 1.02 | 0.64 | 1.11 |
| Fe recovery | RF | 0.65 | 6.288 | 0.91 | 4.196 |
| | ANN | 0.75 | 5.41 | 0.82 | 5.55 |

The RF model for both metal grade and recovery outputs performed better than the respective ANN models with overall $R^2$ values of 0.883 and 0.901, with RMSE values of 4.380 and 3.780, during model training and testing, respectively.

$$R^2 = \frac{\left[\sum_{i=1}^{n}\left(P_i - \overline{P}\right)\left(A_i - \overline{A}\right)\right]^2}{\left[\sum_{i=1}^{n}\left(P_i - \overline{P}\right)^2\left(A_i - \overline{A}\right)^2\right]} \tag{5}$$

$$RMSE = \sqrt{\frac{\sum_{i=1}^{n}(P_i - A_i)^2}{n}} \tag{6}$$

where $n$ = total number of data observations; $P_i$ = predicted value by the model; $A_i$ = actual value in the data; $\overline{P}$ = mean of all predicted values; and $\bar{A}$ is the mean of all the actual values.

The prediction plots for the grades and recoveries of Fe, Pb, and Cu (Figures 10–12, respectively) show that the RF model was able to capture the intrinsic correlations in a reliable manner as indicated by the high $R^2$ values. However, the ANN model was unable to capture the correlations reliably as the $R^2$ values were lower as compared to RF. This is because ANN model uses local search-and-optimization techniques throughout

the training phase, which causes premature convergence [19]. This drawback is often inconsequential—with little-to-no consequence on prediction performance of the models—in datasets wherein the functional relationship between the input variables and output is linear and/or monotonic. However, in the case of froth flotation, the input-output correlations are expected to be complex (presumably highly nonlinear), thus rendering the predictions of ANN models inaccurate—as reflected by the $R^2$ values shown in Table 4. ANN predictions can be improved by combining the model with generic programming algorithms [19] or using the hybrid neural fuzzy models [17] or may be with using deep learning algorithms, which is an advancement of the ANN algorithm [38,57].

## 4. Optimization Studies

The impact of structural characteristics of C-PAM on the flotation of sulfide minerals was previously studied by Monyake and Alagha [8], where statistical analysis (response surface methodology) was used to predict flotation outcomes and for optimization purposes. However, the authors recommended using machine learning models to predict the flotation outcomes and optimize the structural characteristics of C-PAM, as machine learning models could ultimately result in improved and better performance compared to statistical modeling. The results presented in Section 3 showed that the RF model was able to reliably predict the metal grades and recoveries in the flotation concentrates as compared to ANN. An optimization component was developed based on the capability of RF to predict flotation outcomes based on the inputs and desired outputs. The objective of the optimization study was to predict the C-PAM structural characteristics (chitosan degree of deacetylation, $X_1$; chitosan: acrylamide weight ratio, $X_2$ and C-PAM dosage, $X_3$) suitable to produce target grades and recoveries of Fe, Pb, and Cu in the flotation concentrates. The slurry pH, $X_4$ (8), xanthate collector dosage, $X_5$ (450 g/t); sphalerite depressant dosage, $X_6$ (650 g/t); MIBC dosage, $X_7$ (50 g/t); impeller speed, $X_8$ (1250 rpm); and flotation time, $X_9$ (4 min) were kept constant. Based on the value of $X_4$–$X_9$ and targeted metal grades and recoveries, the RF model optimized the value for chitosan degree of deacetylation $X_1$. Then, $X_1$ was an additional input to optimize the chitosan: acrylamide weight ratio, $X_2$. Then, $X_1$ and $X_2$ were used as inputs to optimize the C-PAM dosage $X_3$. The optimization study revealed that 98 g/t of C-PAM synthesized from chitosan of 85% degree of deacetylation at 1:4.5 weight ratio of chitosan: AM was the best structure to produce flotation concentrates with target grades of 3% Fe, 20% Pb, and 2% Cu while recoveries were targeted at 15% Fe, 85% Pb, and 75% Cu, respectively. These target grades and recoveries were optimized previously by the authors in a separate study using statistical methods [8]. The chitosan degree of deacetylation is an important parameter to enhance base metal flotation while depressing pyrite because of the higher content of amine functional group, which was previously reported to have stronger affinity to pyrite compared to base metal sulfides hence enhancing pyrite depression [20,58,59]. This was shown in the XPS study where the amine group chemisorbed on pyrite surfaces had a binding energy shift of +0.11 eV. Grafting acrylamide on chitosan backbone incorporates side chains of acrylamide which are hydrophilic and have stronger affinity to pyrite.

## 5. Conclusions

In this study, random forest (RF) and artificial neural network (ANN) models were developed, trained, and tested to predict the efficiency of in-house synthesized chitosan-polyacrylamide copolymers (C-PAMs) in the depression of pyrite in the bulk flotation process of galena and chalcopyrite hosted in Mississippi Valley Type (MVT) ore. The overall prediction performance of the models was rigorously evaluated based on the coefficient of determination ($R^2$) and the root-mean-square error (RMSE). With the RF model, the overall $R^2$ and RMSE values were 0.88 and 4.38 for the training phase, respectively, and $R^2$ of 0.90 and RMSE of 3.78 for the testing phase. As for the ANN, during the training phase, the overall $R^2$ and RMSE were 0.76 and 4.75, respectively, and during the testing phase, the $R^2$ and RMSE were 0.65 and 5.42, respectively. On the basis of these statistical

parameters, it was clear that, in terms of prediction accuracy, the RF model was superior compared to the ANN model. This was expected and consistent with prior studies that have observed that the prediction performance of the RF model is better than several standalone and ensemble ML models [19,51,60]. The high values of $R^2$ combined with the low values of RMSE strongly suggest that the RF is a reliable tool for predictions of froth flotation efficiency, especially of polymetallic sulfide systems, using experimental process parameters as inputs. However, it is acknowledged that RMSE and $R^2$, by themselves, do not effectively prove that the RF is a better predictor than ANN. This is because the overall prediction performance of any given ML model is affected by various factors, and, therefore, it is complicated to compare different ML models, especially when they are devised and operated by different users. The aforementioned factors include but are not limited to (i) nature and volume of the parent database and its splitting into training and testing sets; (ii) pre-processing (or lack thereof) of the parent database or its derivations (i.e., training and testing sets); (iii) type and number of statistical parameters (e.g., $R^2$ and RMSE) used for assessment of prediction performance; and (iv) techniques used to optimize the hyper-parameters of the ML models. Additionally, this work focused on the use of using environmentally friendly reagents (C-PAM), which were synthesized in-house for the depression of pyrite. C-PAM was found to strongly adsorb on pyrite as compared to galena and chalcopyrite through zeta potential, XPS, and adsorption density measurements. This finding could potentially prove to be relevant to the industrial flotation of a sulfide ore where galena and chalcopyrite are floated in bulk while pyrite is depressed in the first stage of flotation. The adsorption of C-PAM on pyrite was suggested to be through the chemisorption of the amine and amide groups of C-PAM.

**Supplementary Materials:** The following supporting information can be downloaded at: https://www.mdpi.com/article/10.3390/colloids7020041/s1, Figure S1: X-Ray Diffraction spectra of model sulfide minerals to illustrate purity of (a) galena; (b) chalcopyrite; (c) pyrite and (d) sphalerite; Figure S2: 1H NMR spectra of chitosan, PAM and C-PAMs synthesized from chitosan of 85% DD at different weight ratios of chitosan: AM; Figure S3: ATR-FTIR spectra of (a) chitosan of 85% DD; (b) PAM; (c) C-PAM of 1:3 ratio; (d) C-PAM of 1:5 ratio and (e) C-PAM of 1:7 ratio; Figure S4: Variables importance of the random forest model inputs for the grades of (a) lead; (b) copper and (c) iron; Figure S5: Variables importance of the random forest model inputs for the recoveries of (a) lead; (b) copper and (c) iron; Table S1: ML models inputs and outputs for experiments 1–35; Table S2: ML models inputs and outputs for experiments 36–70; Table S3: ML models inputs and outputs for experiments 71–105; Table S4: ML models inputs and outputs for experiments 106–130.

**Author Contributions:** Conceptualization, K.M. and L.A.; methodology, K.M.; software, D.A. and T.H.; validation, D.A. and T.H.; formal analysis, K.M. and L.A.; investigation, K.M. and L.A.; resources, L.A. and A.K.; data curation, T.H., D.A., and K.M.; writing—original draft preparation, K.M., T.H., and D.A.; writing—review and editing, L.A.; visualization, T.H. and D.A.; supervision, A.K. and L.A.; project administration, A.K. and L.A.; funding acquisition, L.A. All authors have read and agreed to the published version of the manuscript.

**Funding:** This research received no external funding.

**Conflicts of Interest:** The authors declare no conflict of interest.

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
