# Peer review of "Experimental and Machine Learning Studies on Chitosan-Polyacrylamide Copolymers for Selective Separation of Metal Sulfides in the Froth Flotation Process"

_colloids, doi:10.3390/colloids7020041_

Round 1

Reviewer 1 Report

Reviews about the manuscript named "Experimental and Machine Learning Studies on Chitosan-Polyacrylamide Copolymers for Selective Separation of Metal Sulfides in Froth Flotation Process" are given below:

·   In Section 2.1, it is stated that the purity of the minerals used in the experiments is above 85%. This should be proven by XRD and/or chemical analysis.

·        Specifications of reagents used in experimental studies need to be given (brand, manufacturer, etc.)

·         pH is one of the most important parameters in experiments where galena and chalcopyrite are floated while sphalerite and pyrite are suppressed. I couldn't see any data about this in Table 1.

·         Flotation data should be given in more detail. As can be seen from Table 2, the floating of galena and sphalerite is quite low.

·         The characteristics of the photoelectron spectrometer should be specified.

·         Based on zeta potential measurements between pH:8-12, it is indicated that pyrite can be suppressed with reference to Figure 3. However, the same effect is present in galena and chalcopyrite at the same pH values. In order to use the difference expression, the values must be quite different from each other.

·         In the material and method, sphalerite was mentioned among the sulfur minerals contained in the ore. However, no measurements were made on sphalerite.

·         The "R" values shown in Figures 9, 10, 11, 14, 15 and 16 will probably be "R2".

·         Were the flotation experiments mentioned in the article performed in a single step or in multiples? It must be specified.

·         It is stated that the 20% Pb content specified in the "Optimization Process" section should be reached. Could this value be reached with these data?

·         Line 602 has reference 61. It doesn't show up in the references section.

Author Response

Responses to Reviewer's comments can be found in the attached file.

Reviewer 2 Report

This manuscript proposed two machine learning (ML) models, i.e., artificial neural network (ANN) and random forests (RF), to predict the depression of pyrite and flotation of galena with reasonably good performance, while fundamental investigations on the surface chemistry of C-PAMs at mineral-water interface were conducted to understand the behavior and mechanism of the flotation process. The comments and suggestions are listed as follows.

1, The authors used two machine learning models (RF and ANN) to predict the grade and recovery of Pb, Fe, Cu, and Zn, with the results showing that RF has better performances. There are nine input variables. Could the authors explain the importance of each variable of the RF model?

2, More information is recommended to be provided for the samples of minerals, just like the grade of Pb, Fe, Cu, and Zn in samples. Also, the process flow diagram of the flotation process is to be described.

3, What is the main reason for the authors to choose 100 trees in the forest and 4 splits at each node?

4, The full form of “ML” should be provided when the abbreviation appears in the text at first use.

5, Wherein Lines 170 and 202, should be lowercase.

Author Response

Our responses to reviewer's comments can be found in the attached file.

Round 2

Reviewer 1 Report

The authors have made the requested changes. It can be published as it is only after placing the specified supplementary in the text.